# meiRNA, A Polyvalent Player in Fission Yeast Meiosis

**DOI:** 10.3390/ncrna5030045

**Published:** 2019-09-17

**Authors:** Akira Yamashita

**Affiliations:** 1National Institute for Basic Biology, Nishigonaka 38, Myodaiji, Okazaki 444-8585, Japan; ymst@nibb.ac.jp; Tel.: +81-564-557-512; 2Center for Novel Science Initiatives, National Institutes of Natural Sciences, Nishigonaka 38, Myodaiji, Okazaki 444-8585, Japan; 3Department of Basic Biology, School of Life Science, SOKENDAI (The Graduate University for Advanced Studies), Nishigonaka 38, Myodaiji, Okazaki 444-8585, Japan

**Keywords:** non-coding RNA, nuclear body, meiosis, fission yeast, gene expression, chromosome pairing

## Abstract

A growing number of recent studies have revealed that non-coding RNAs play a wide variety of roles beyond expectation. A lot of non-coding RNAs have been shown to function by forming intracellular structures either in the nucleus or the cytoplasm. In the fission yeast *Schizosaccharomyces pombe*, a non-coding RNA termed meiRNA has been shown to play multiple vital roles in the course of meiosis. meiRNA is tethered to its genetic locus after transcription and forms a peculiar intranuclear dot structure. It ensures stable expression of meiotic genes in cooperation with an RNA-binding protein Mei2. Chromosome-associated meiRNA also facilitates recognition of homologous chromosome loci and induces robust pairing. In this review, the quarter-century history of meiRNA, from its identification to functional characterization, will be outlined.

## 1. A Long Non-Coding RNA Essential for the Progression of Meiosis in Fission Yeast

It is nearly impossible to avoid issues relating to non-coding RNAs in biology nowadays. Non-coding RNAs play crucial roles in a wide range of biological processes. In particular, an increasing number of non-coding RNAs have been shown to be active in gene expression regulation. Participation of non-coding RNAs in gene expression makes great contribution to enable fine-tuned regulation to adapt to environmental changes. Furthermore, it has been demonstrated that non-coding RNAs function in diverse cellular contexts beyond gene expression [1,2].

In the fission yeast *Schizosaccharomyces pombe*, a unicellular eukaryotic model organism, a long non-coding RNA termed meiRNA acts as an essential factor for meiosis. *S. pombe* cells grow preferentially in the haploid state. When starved for nutrients, *S. pombe* cells conjugate with cells of the opposite mating type and form zygotes. Resultant diploid zygotes then undergo meiosis and eventually form spores that are tolerant to stress conditions including starvation [3]. meiRNA was initially identified a quarter-century ago in the screening for high copy suppressors of a conditional meiotic mutant, *mei2*, which was defective in the initiation of the first meiotic division (meiosis I) [4]. Removal of meiRNA from *S. pombe* cells by deleting the *sme2* gene that encodes meiRNA also results in meiotic cell cycle arrest before meiosis I, indicating an essential role of meiRNA in the induction of meiosis I. meiRNA was originally reported to be about 0.5 kb polyadenylated transcripts [4]. meiRNA has lately been shown to have two types of isoforms, meiRNA-S (about 0.5 kb) and meiRNA-L (about 1.5 kb) (Figure 1) [5]. Whereas meiRNA-L is shown to have full activity to carry forward meiosis, the function, if any, of meiRNA-S remains unknown [6,7].

## 2. Binding Partner of meiRNA

Over-production of meiRNA suppresses the meiotic defect of the *mei2* mutant. The *mei2* gene encodes an RRM-type RNA binding protein. This evokes the possibility of meiRNA directly interacting with Mei2 protein. It has been actually demonstrated that Mei2 binds to the 5′ region of meiRNA-L, corresponding to meiRNA-S [4,6]. meiRNA also binds to another RNA-binding protein, Mmi1, through the 3′ region, as mentioned below. Interaction of meiRNA with Mei2 and Mmi1 has also been confirmed by several genome-wide high-throughput approaches [8,9,10,11].

Mei2 is a key player in the initiation and progression of meiosis in *S. pombe* [12]. Mei2 is inactivated by phosphorylation in mitotically growing cells [13]. Forced expression of an unphosphorylated active form of Mei2 leads to induction of the whole process of meiosis even in haploid cells under nutrient rich conditions, indicating that Mei2 is a molecular switch to turn on meiosis [13]. Mei2 is also removed from mitotically growing cells through the ubiquitin-proteasome pathway [14,15]. An intriguing and unanswered question is how Mei2 induces meiosis.

After inducing meiosis, Mei2 has another function of inducing meiosis I. This was demonstrated using a temperature-sensitive *mei2* mutant that is able to initiate meiosis, but does not undergo meiosis I [4]. The following observations led to the conclusion that Mei2 and meiRNA play an essential role in meiosis I as an RNA-protein complex: The meiosis I defect in the *mei2* mutant is suppressed by over-expression of meiRNA; cells lacking meiRNA cannot enter meiosis I; meiRNA directly associates with Mei2 both in vivo and in vitro [4].

## 3. Mei2 Dot

Microscopic observations revealed that the Mei2–meiRNA complex functions by forming a nuclear dot structure, known as Mei2 dot [13,16]. In cells lacking meiRNA or in cells expressing a mutant form of Mei2 that has lost its ability to bind to meiRNA, the Mei2 dot is not formed, indicating that interaction between Mei2 and meiRNA is crucial for the assembly of the Mei2 dot [16].

During the premeiotic S phase and meiotic prophase, the *S. pombe* nucleus undergoes oscillatory movement between the two poles of the cell [17]. This dynamic nuclear movement is called “horsetail” movement due to its shape, and it facilitates homologous chromosome pairing, which is essential for proper meiotic chromosome segregation. During the horsetail movement, the nucleus is led by a spindle pole body (SPB), a centrosome equivalent structure, to which all the telomeres are clustered, forming a bouquet chromosome arrangement [17,18]. The Mei2 dot was observed at a fixed position near the SPB in the oscillating horsetail nucleus, suggesting its association with a specific region of the chromosome (Figure 2a) [13,16]. Another study using chromosome translocation has revealed that the Mei2 dot associates with the *sme2* locus on chromosome II, which encodes meiRNA [19]. meiRNA has also been demonstrated to localize to its genetic locus, *sme2* (Figure 2b) [6,16,19]. Mei2 is a shuttling protein between the nucleus and cytoplasm [20]. From these observations, it is hypothesized that meiRNA is tethered to the *sme2* locus, encoding itself, after transcription by an unclear mechanism, and then catches the shuttling Mei2 through direct interaction, to form the Mei2 dot.

## 4. Molecular Function of Mei2 Dot

The next natural question is about the molecular function of the Mei2 dot. To get a clear answer, we have had to wait for identification and characterization of another key player, Mmi1.

The first step was a high-copy number suppressor screening for the meiotic arrest of cells lacking meiRNA [21]. Although no common feature was found among the isolated suppressors, some of them were expressed exclusively during meiosis. These included *mei4*, *ssm4*, *rec8*, and *spo5*, all of which encode factors required for proper meiosis progression. Intriguingly, *mei4* transcripts did not accumulate in mitotically growing cells, even when they were forced to express from a strong constitutive promoter [22]. A specific region on *mei4* transcripts was shown to be responsible for their elimination from mitotic cells. When this region was fused to a reporter gene such as GFP, the resultant chimeric gene was expressed exclusively in meiotic cells. Similar regions responsible for meiosis-specific expression were found in *ssm4, rec8,* and *spo5*. These cis-acting regions were designated as determinant of selective removal (DSR). DSR activity was demonstrated to be exhibited by the enrichment of the hexanucleotide UNAAC motif [5,8]. DSR-carrying transcripts are not limited to the four genes mentioned above, and it is assumed that dozens of transcripts have DSR.

Mmi1, a key factor for DSR-dependent meiotic transcript elimination, was isolated by genetic screening [22]. Mmi1 is a YTH-type RNA binding protein and interacts with DSR directly, by recognizing the UNAAAC motif [5,22]. While YTH-RNA-binding proteins generally recognize N6-methyladenosine (m6A)-containing RNAs [23], Mmi1 binds to its target RNAs in a unique manner and does not recognize m6A [24,25,26]. When DSR-carrying transcripts are recognized by Mmi1, RNA degradation by the nuclear exosome, a highly conserved 3′-5′ exonuclease complex, is induced [22,27]. This nuclear exosome-mediated RNA degradation requires polyadenylation by a canonical poly(A) polymerase [27]. Mmi1 also suppresses expression of a subset of its target genes by inducing facultative heterochromatin assembly at these loci [8,28,29]. Moreover, Mmi1 regulates transcriptional termination of its target genes [10,30,31] and prevents nuclear export of its target transcripts [32]. Mmi1 associates with a nuclear protein complex called MTREC (Mtl1-Red1 core) composed of a zinc-finger protein, Red1, and an Mtr4-like helicase, Mtl1 [33,34]. MTREC is suggested to be a counterpart of PAXT (poly(A) tail exosome targeting), which plays a crucial role in the selective elimination of polyadenylated transcripts in the nucleus of human cells [35]. MTREC is essential for Mmi1-mediated RNA degradation and facultative heterochromatin assembly [33,34]. Mmi1 also associates with a conserved multiprotein complex, Ccr4-Not [36,37,38,39]. It is suggested that Not4/Mot2 ubiquitin ligase subunit of the Ccr4-Not complex is recruited to Mei2 by Mmi1 and suppresses expression of DSR-containing transcripts including meiRNA by limiting the Mei2 accumulation in mitotically growing cells [39]. These Mmi1-mediated multi-layered regulations on meiotic genes may reflect the risk of their mistimed expression, which has been demonstrated to cause chromosome segregation errors [40].

Mmi1 is crucial for mitotic growth, but becomes an obstacle for meiosis, since it suppresses expression of meiotic genes. Thus, the Mmi1-DSR system should be blocked during meiosis. In this regard, the function of the Mei2 dot was found out to be inhibition of Mmi1 [22]. In mitotically growing cells, Mmi1 forms several nuclear foci. Many factors involved in the Mmi1-DSR system, including the nuclear exosome, localize to the Mmi1 nuclear foci [27,34,38,41,42,43]. DSR transcripts also localize to the Mmi1 foci [32]. When cells initiate meiosis, the Mmi1 foci converge on a single focus [22]. Importantly, the Mmi1 convergence point coincides with the Mei2 dot. Furthermore, Mmi1 does not converge in *mei2* or *sme2* deletion mutant cells both of which fail to form the Mei2 dot. From these observations, it has been thought that Mmi1 may be sequestered and inactivated by the Mei2 dot. Physical interaction of Mmi1 with Mei2 and with meiRNA supports this hypothesis [6,22]. Moreover, consistent with the hypothesis, the meiotic arrest of cells lacking the Mei2 dot could be suppressed by a hypomorphic *mmi1* mutation [22]. As mentioned before, the meiosis arrest phenotype in cells lacking meiRNA is suppressed by overexpression of DSR-carrying transcripts [5,21]. Based on the hypothesis, this is rational because excess target transcripts may sequester Mmi1.

meiRNA carries numerous copies of the DSR motif in its 3′ region (Figure 1) and binds to Mmi1 through the 3′ region [6]. Moreover, meiRNA is degraded by the Mmi1-mediated degradation system in mitotically growing cells and is stably expressed only in meiotic cells [5,6]. Expression of meiRNA is upregulated by a transcription factor, Ste11, upon nutrient starvation [4,44]. From these observations, it is suggested that meiRNA serves as a decoy to lure Mmi1. meiRNA is indeed able to reduce the activity of Mmi1, while complete inactivation of Mmi1 might require Mei2 [6]. In mitotically growing cells, overexpression of meiRNA induces ectopic expression of Mmi1-target DSR transcripts in *mmi1* mutant cells but not in wild-type cells. Meanwhile, in mutant cells lacking the 5′ region of meiRNA, to which Mei2 binds, DSR transcripts are expressed upon nutrient starvation, although lower than in wild-type cells. The mutant cells lacking the 5′ region are able to undergo meiosis [6], suggesting that the Mmi1 function may be repressed through an unknown mechanism in the course of meiosis and that inactivation of Mmi1 may be accomplished by meiRNA, independent of the binding of Mei2.

## 5. Contribution of meiRNA in Homologous Chromosome Pairing

It has been demonstrated that meiRNA plays a distinct role other than meiotic gene expression regulation [7]. In vivo live-cell imaging has revealed that the two *sme2* loci on homologous chromosomes, which encode meiRNA, pair robustly in the early stage of meiotic prophase. When translocated to another site, the *sme2* locus induces robust pairing there. Furthermore, the *sme2* locus is able to facilitate ectopic pairing between nonhomologous chromosome sites when both sites carry the locus. The robust pairing at the *sme2* locus depends on transcription of meiRNA. Dampening of meiRNA transcription at one *sme2* locus of two homologous chromosomes impairs the robust early pairing. From these observations, it is thought that meiRNA stays at its genetic locus and triggers recognition and interaction of the meiRNA-associated site on homologous chromosomes through unknown mechanisms. A 5′ region of meiRNA, corresponding to meiRNA-S, is dispensable for induction of robust pairing. meiRNA binds to Mei2 through this 5′ region [4,6], indicating that the association of Mei2 with meiRNA is not necessary for the robust pairing [7]. Chromosome tethering of meiRNA at the *sme2* locus requires Mmi1, to which meiRNA binds through the 3′ region [6]. However, involvement of Mmi1 in chromosome pairing remains to be proven.

## 6. Outlook

Multiple roles of meiRNA during meiosis have been demonstrated, namely, the regulation of meiotic gene expression and induction of homologous chromosome pairing. meiRNA might be one of the most characterized non-coding RNAs, but there remain unanswered questions. For instance, it is unclear how meiRNA is tethered to its genetic locus. It has been shown that Mmi1 is essential for chromosomal tethering of meiRNA [6], but the precise mechanisms are yet to be clarified. Further studies are also required to understand the molecular mechanisms of meiRNA-mediated pairing. Moreover, yet another intriguing question is whether RNA-driven pairing is observed at other loci. Several non-coding RNAs have been demonstrated to be upregulated during meiosis [45], although their molecular function is unknown. Future studies will tell us how meiRNA functions, in greater detail, and shed light on the generality of such non-coding RNA-mediated regulations.

## Figures and Tables

**Figure 1 ncrna-05-00045-f001:**
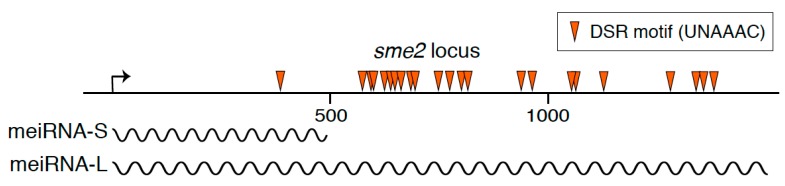
A schematic diagram of the *sme2* locus on chromosome II, which encodes meiRNA. Wavy lines indicate two isoforms of the meiRNA transcripts, namely meiRNA-S and meiRNA-L. Arrowheads indicate the determinant of selective removal (DSR) motif, UNAAAC, which is recognized by Mmi1.

**Figure 2 ncrna-05-00045-f002:**
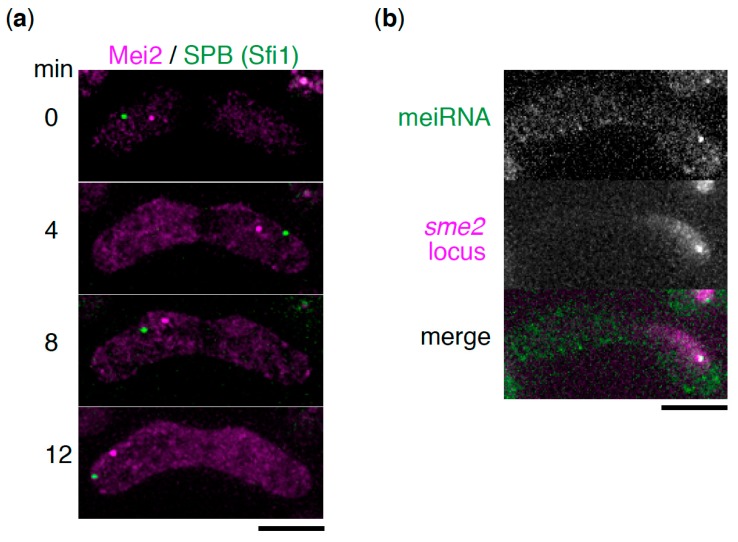
meiRNA forms a chromosome-associated dot structure in cooperation with its binding partner, Mei2. (**a**) Time-lapse images of Mei2 (magenta) and the spindle pole body (SPB) (Sfi1, green) during the horsetail movement. Scale bar: 5 µm. (**b**) Localization of meiRNA. meiRNA was visualized by the MS2 system, in which MS2-loop-tagged meiRNA and MS2-YFP were expressed (green) [6]. The *sme2* locus was marked by inserting *lacO* repeats and expressing 4xmCherry-LacI-NLS (magenta). Scale bar: 5 µm.

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
