# Peer review of "meiRNA, A Polyvalent Player in Fission Yeast Meiosis"

_ncrna, 2019, doi:10.3390/ncrna5030045_

Round 1

Reviewer 1 Report

This review is contributed by Dr Akira Yamashita, an internationally recognized leader in the field of the control of meiosis entry by meiRNA in S. pombe.

This review is well written and briefly presents the state of the art on meiRNA and also highlight the still opened questions in the field. 

It should be of general interest for the readership of non-coding RNA.

I only have a minor comment as in section 5, the several occurrences of "paring" should be replaced by "pairing".

Author Response

I apologize for the overlook; I have corrected the text in section 5 and Abstract (line 19).

Reviewer 2 Report

The review covers 25 years of research on the fission yeast meiosis-specific long noncoding RNA meiRNA, from identification to functional characterization. The manuscript is divided in 5 main paragraphs describing how meiRNA and its binding partners, the RNA-binding proteins Mei2 and Mmi1, were identified and what is currently known about their role in meiosis.

The review is of interest to scientists working on the mechanisms involved in the control of sexual differentiation in S. pombe, and more generally on lncRNA biology. However, prior to publication, additional concepts and citations currently missing should be discussed/added:

It is mentioned that meiRNA is degraded by the Mmi1 system during vegetative growth (lane 138) but whether transcriptional and/or post-transcriptional mechanisms ensure the build-up in meiRNA levels in meiosis is not sufficiently discussed. This would help better understanding how meiRNA efficiently lures Mmi1. Mei2 binds to the 5’ part of meiRNA, yet cells expressing a variant of meiRNA lacking this region are still able to proceed through meiosis (Shichino et al., 2014). This might suggest that the binding of Mei2 to meiRNA is not mandatory for meiosis. The author should discuss this point. It is mentioned in section 2 (lane 56) that Mei2 is regulated by phosphorylation but references are missing. In addition, Mei2 is also ubiquitinylated. Papers exploring this aspect could be cited (Kitamura et al., 2001; Otsubo et al., 2014; Simonetti et al., 2017). The binding of Mmi1 and Mei2 to meiRNA during mitosis and meiosis has been validated by genomewide approaches (Kilchert et al., 2015; Touat-Todeschini et al., 2017; Mukherjee et. al., 2018). Mmi1 associates with two effectors complexes, MTREC and Ccr4-Not, neither of which is mentioned in the review. It is important because MTREC functions with Mmi1 to mediate meiRNA degradation (Lee et al., 2013). Likewise, Mmi1 associates with Ccr4-Not to degrade Mei2, which is required to maintain low levels of meiRNA in vegetative cells (Simonetti et al., 2017). In section 4, the details about the functions of genes containing DSR (lanes 97 to 101) are not mandatory to appreciate meiRNA biology.

Typos:

lane 42: the sentence should end after “(Figure 1) [5]” lanes 148, 150, 154: “pairing”, not “paring”

Author Response

Comment 1: It is mentioned that meiRNA is degraded by the Mmi1 system during vegetative growth (lane 138) but whether transcriptional and/or post-transcriptional mechanisms ensure the build-up in meiRNA levels in meiosis is not sufficiently discussed. This would help better understanding how meiRNA efficiently lures Mmi1.

Response: According to the reviewer’s suggestion, I have modified the text and added a sentence to mention transcriptional and post-transcriptional regulation of meiRNA (page 4, line 145).

Comment 2: Mei2 binds to the 5’ part of meiRNA, yet cells expressing a variant of meiRNA lacking this region are still able to proceed through meiosis (Shichino et al., 2014). This might suggest that the binding of Mei2 to meiRNA is not mandatory for meiosis. The author should discuss this point.

Response: I appreciate the reviewer’s comment. Accordingly, I have added sentences in the revised manuscript (page 4, line 148).

Comment 3: It is mentioned in section 2 (lane 56) that Mei2 is regulated by phosphorylation but references are missing. In addition, Mei2 is also ubiquitinylated. Papers exploring this aspect could be cited (Kitamura et al., 2001; Otsubo et al., 2014; Simonetti et al., 2017).

Response: As per the reviewer’s suggestion, I have cited the paper showing that Mei2 is inactivated by phosphorylation (page 2, line 57). I have also added a sentence referring to degradation of Mei2 by the ubiquitin-proteasome pathway and cited the papers (page 2, line 60).

Comment 4: The binding of Mmi1 and Mei2 to meiRNA during mitosis and meiosis has been validated by genomewide approaches (Kilchert et al., 2015; Touat-Todeschini et al., 2017; Mukherjee et. al., 2018).

Response: According to the reviewer’s comment, I have added a sentence to mention this point (page 2, line 54).

Comment 5: Mmi1 associates with two effectors complexes, MTREC and Ccr4-Not, neither of which is mentioned in the review. It is important because MTREC functions with Mmi1 to mediate meiRNA degradation (Lee et al., 2013). Likewise, Mmi1 associates with Ccr4- Not to degrade Mei2, which is required to maintain low levels of meiRNA in vegetative cells (Simonetti et al., 2017).

Response: I would like to thank the reviewer for pointing this out. I have added sentences to explain roles of MTREC and Ccr4-Not in Mmi1-mediated regulation (page 4, line 118).

Comment 6: In section 4, the details about the functions of genes containing DSR (lanes 97 to 101) are not mandatory to appreciate meiRNA biology.

Response: I agree with the reviewer. I have removed the details of DSR-containing genes (page 3, line 98).

Typos: lane 42: the sentence should end after “(Figure 1) [5]” lanes 148, 150, 154: “pairing”, not “paring”

Response: I have corrected the text now (page 1, line 42; page 1, line 19; page 4, line 156, line 164;page 5, line 168).